# Impact of COVID-19 on Electricity Demand: Deriving Minimum States of System Health for Studies on Resilience

Smruti Manjunath [1,*], Madhura Yeligeti [1], Maria Fyta [2], Jannik Haas [1,3] and Hans-Christian Gils [1,*]

1. Institute of Networked Energy Systems, German Aerospace Center (DLR), 70563 Stuttgart, Germany; Madhura.Yeligeti@dlr.de
2. Institute for Computational Physics, University of Stuttgart, 70569 Stuttgart, Germany; mfyta@icp.uni-stuttgart.de
3. Department of Stochastic Simulation and Safety Research for Hydrosystems (IWS/SC SimTech), University of Stuttgart, 70569 Stuttgart, Germany; jannik.haas@iws.uni-stuttgart.de
* Correspondence: Smruti.Manjuanth@dlr.de (S.M.); Hans-Christian.Gils@dlr.de (H.-C.G.)

**Abstract:** To assess the resilience of energy systems, i.e., the ability to recover after an unexpected shock, the system's minimum state of service is a key input. Quantitative descriptions of such states are inherently elusive. The measures adopted by governments to contain COVID-19 have provided empirical data, which may serve as a proxy for such states of minimum service. Here, we systematize the impact of the adopted COVID-19 measures on the electricity demand. We classify the measures into three phases of increasing stringency, ranging from working from home to soft and full lockdowns, for four major electricity consuming countries of Europe. We use readily accessible data from the European Network of Transmission System Operators for Electricity as a basis. For each country and phase, we derive representative daily load profiles with hourly resolution obtained by k-medoids clustering. The analysis could unravel the influence of the different measures to the energy consumption and the differences among the four countries. It is observed that the daily peak load is considerably flattened and the total electricity consumption decreases by up to 30% under the circumstances brought about by the COVID-19 restrictions. These demand profiles are useful for the energy planning community, especially when designing future electricity systems with a focus on system resilience and a more digitalised society in terms of working from home.

**Dataset:** https://doi.org/10.6084/m9.figshare.c.5513850.v1.

**Dataset License:** CC0

**Keywords:** energy modelling; energy planning; COVID-19; energy demand; resilience; remote-working





## 1. Introduction

Assessing the resilience of energy systems, i.e., the ability to recover after an unexpected shock [1], is becoming more relevant in the context of rapidly changing energy systems and climate-change-related disasters [2]. Studies on resilience commonly measure how disasters impact a system's ability to serve the energy demand and how quickly the system can be restored to a minimum level of service [3]. However, precise quantitative descriptions of such minimum states of service are virtually non-existent [4]. These are inherently difficult to obtain as extreme disasters are, by definition, rare, which is why this task is commonly reduced to imperfect expert knowledge [4,5]. Recently, the measures adopted by governments in Europe to limit the spread of COVID-19 have impacted the energy demand and provide a rich numeric dataset, which can serve as proxy for such states of minimum service.

Furthermore, our changing society will impact future electricity demand profiles [6]. Adequate consideration of these changing load profiles is an essential element in planning future energy systems [7]. Digital technologies revolutionize not only consumer and mobility behaviour but also industrial production and service activities. Since the effects of COVID-19 measures like behavioural and occupational changes are comparable to those induced by a greater proliferation of digital technologies, the resulting load profiles can serve as a first approximation of this transformation.

The impact of the measures designed to curtail the spread of COVID-19 on electricity demand has been analysed in previous works. In [8], the authors define a demand variation index to compare the change in electricity consumption in some European countries with severe restrictive measures to that in two European countries with less restrictive measures. In [9], the dynamics of the electricity demand within a week prior to and post-COVID-19 are explored, for the province of Ontario in Canada. Reference [10] analyses the electricity demand for three regions in Jordan with a focus on eliminating the time series correlation, trends and seasonality impact so as to cover only the pandemic's impacts. In [11], the authors provide an econometric approach to quantify the changes in national electricity demand for Germany, France, and Great Britain by using the specific lockdown periods and the number of active cases as instrumental variables in regression analysis. Reference [12] studies the impact of COVID-19 shutdowns on the level of electricity demand in Europe and its weekly pattern using high-dimensional regression techniques.

However, the focus of these analyses is limited to addressing the structural and behavioural changes and their influence on the electricity demand and load forecasting. They do not categorise the restrictions into phases of varying degrees of stringency, nor are daily load profiles which are representative of each of these phases derived. Furthermore, the change in electricity demand for each phase is not systematised in these works. Additionally, they analyse the change in demand by comparing the demand in 2020 with only one reference year rather than deriving a reference demand profile representative of several years, which can potentially bias the results. Finally, they lack the open provision of their underlying research datasets.

In this work, we aim to address these gaps in the literature by deriving representative electricity demand profiles resulting from the different measures of containing COVID-19 for the four largest electricity consumers of the European Union: Germany, Italy, Spain, and France [13]. These profiles serve as a data basis for describing the minimum operational state of the electricity system pertinent to studies on system resilience. Furthermore, these profiles can also be used to describe future energy demands of a society with a higher adoption of remote working.

We identify and categorise the measures adopted by the respective governments from February 2020 to May 2020 into three broad phases in increasing order of severity—work-from-home (WFH), soft lockdown (SLD), and full lockdown (FLD)—to analyze the electricity demand of each country under these phases. We use readily accessible data from the European Network of Transmission System Operators for Electricity (ENTSO-E) to derive representative daily load profiles with an hourly resolution for each of these phases and for each country considered. This involved identifying the most representative value for the load at each hour as the medoid of a cluster [14]. These data were further processed to systematise the change in demand under each phase for all countries. Weekdays and weekends are considered separately.

The resulting dataset contains the aforementioned representative daily load profiles. From this dataset, it becomes clear that not only does the overall electricity consumption decrease by around 15 to 30%, but also the shape of the load profiles is significantly influenced by the behavioral changes triggered by the COVID-19 measures. Noteworthy is that the classical working-hour peak is flattened considerably.

The derived datasets can be valuable to the energy community in the following ways:

1.  The profiles can serve as direct inputs into energy system models, especially in assessments focused on future changes in electricity load patterns [6,7]. This includes

the assessment of increasing work from home on the balancing needs for future energy systems [15].

2. For studies on system resilience, the data reflecting the impact of the full lockdown can be interpreted as the first state to be attained in the recovery phase after an extreme event and help identify minimum states of system service [16].
3. The amount and profile of electricity demand in the minimum state of system service can be used to derive tailored options for load shedding in blackout situations and for more efficient energy management [17,18].
4. The systematization of the decrease in demand helps to forecast or approximate the electricity demand for countries with similar energy consumption patterns, whenever the load is not available.
5. This dataset can also be helpful for making bids in electricity markets in the future, when remote working or lockdown situations are expected [19].

The remaining sections are structured as follows: Section 2 details the methods adopted to obtain the datasets, followed by Section 3, which provides a concise description of the datasets published. Next, Section 4 discusses the interpretation of the datasets from the perspective of previous studies and highlights future research directions. Finally, Section 5 addresses some limitations of the data and inconsistencies in the expected trend of electricity demand.

## 2. Methods

First, to set the scope for our study, Section 2.1 introduces and classifies the restrictions imposed by governments in spring 2020. Following that, the data collection process in explained in Section 2.2. The selection of reference data for the interpretation of the generated data sets is described in Section 2.3. Finally, Section 2.4 entails a step-by-step description of data processing.

### 2.1. Background Information and Definitions

In Section 2.1.1, we define the categories of measures that were adopted against the spread of COVID-19. Next, in Section 2.1.2, country-specific restrictions implemented from February to May 2020 in Germany, France, Italy, and Spain are delineated and classified into the aforementioned categories. These definitions and the subsequent categorisation serve as the first steps in the methodology and allow for a precise determination of the duration of each category or phase.

### 2.1.1. Definition of Restriction Categories

The behavioural and occupational changes induced by COVID-19 measures, like compulsion to stay at home, limitations on professional and social activities and drastic reductions in services and industry are expected to change the electricity consumption. In order to analyse this change under different scenarios, we classify the specific measures adopted by governments into three categories of increasing intensity, which we define below.

The WFH category identified by us collectively describes those individual measures adopted by governments and/or industries, the nature of which typically instruct citizens (employees, clients, students, etc.) to stay at home and engage in remote working. The SLD category constitutes those measures whose scope is indicative of closure of public facilities and spaces and prohibition of public activities. This definition is similar to the soft lockdown phase described in [20] and akin to soft scenarios introduced in [21] and mild lockdown in [22]. In addition to all measures under the WFH and SLD categories, the FLD category refers to severe restrictions to movement, physical contact, and halting of production in industries. The FLD category, also connoted as hard lockdown in [20,22], reflects the minimum operating state of the economy where only critical industries and businesses are operational while all non-essential businesses are closed. The restrictions in these categories, as well as their relation, are classified in Figure 1. It is important to

note that definitions and durations of these measures vary for each country as described in the following.

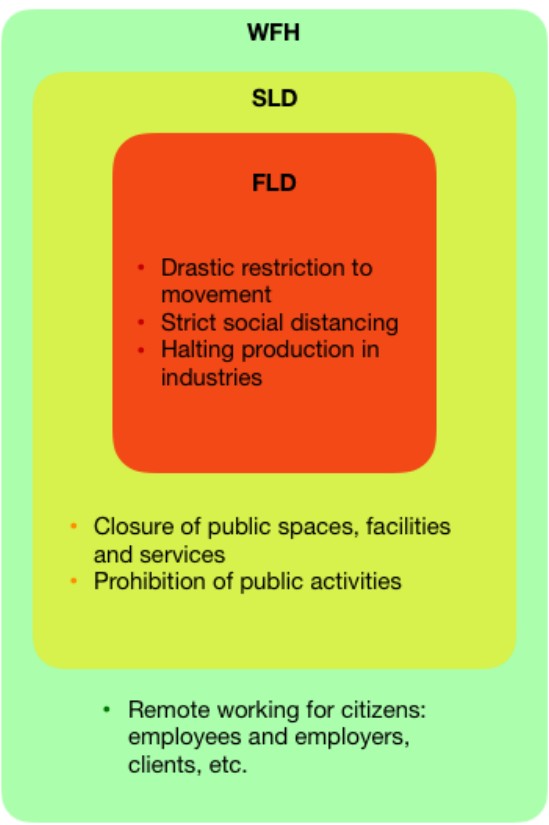

**Figure 1.** Definition and classification of categories for the restrictions taken to prevent the spread of COVID-19 in spring-2020. The red, yellow, and green boxes represent the Full Lockdown (FLD), Soft Lockdown (SLD), and Work-from-Home (WFH) phases, respectively. The FLD phase includes the measures adopted during the SLD phase, which in turn includes those adopted during the WFH period.

2.1.2. Country Specific Implementation of the Restrictions

This section details the classification of the individual measured adopted by the governments of Germany, Italy, Spain, and France into the aforementioned WFH, SLD, and FLD phases. The focus of this study is the time period from February to May 2020.

While Germany never explicitly issued a WFH order to companies and employers, it recommended the "home office" set-up and published FAQs related to remote working as early as the 9th of March [23,24]. The German health minister, in an interview on 11th March, made a recommendation to work remotely wherever possible [25]. Furthermore, references [26,27] are suggestive of German employees transitioning to remote working. The SLD in Germany includes (but is not just limited to) a prohibition on travelling in coaches, attending religious meetings, visiting playgrounds or engaging in tourism, along with a closure of all non-essential shops [28,29]. This period lasted from the 17th to 20th of March. In the FLD phase (23rd March through 30th April), even stricter restrictions were imposed including strict social distancing, closure of many more services and restaurants and an extension of the closure of schools in the country [30,31].

In Italy, major companies allowed employees to work remotely [32,33]. However, the region of Lombardy in Northern Italy, which was the epicentre of the virus outbreak was already experiencing SLD-like measures with closure of schools and cancellation of public events including religious services and sporting events, suspension of activities in universities and educational trips [34–36], suspension of sporting activities in Lombardy in Veneto [37] and University of Bologna setting up a remote teaching project [38]. This period,

from the 24th to 28th of February and classified as the WFH phase in this study, actually saw more region-specific measures being announced rather than nation-wide measures. The SLD in Italy began on the 2nd of March, with the classification of the country into three zones, aimed at curtailing the spread out of the outbreak. In this period, a nation-wide shutdown of schools and universities were imposed [39], as well as a ruling from the government, which meant that all sporting events in the nation could only be played behind closed doors [40]. The FLD in Italy was an extension of the measures adopted for the hard-hit regions of Northern Italy to the national scale [41], and a tightening of the lockdown, which included the closing down of all commercial and retail businesses except those offering essential services [42]. The nation-wide lockdown, lasting all through April until the 1st of May, also included tighter regulations on free movement [43] and even a halt on all non-essential production [44].

In Spain, similar to Germany, no official order to shift to remote working was announced, but the Ministry of Employment published a practical guide. This included recommendations to adopt certain measures such as teleworking and to even halt company activities [45]. This WFH period lasted from the 9th through 13th of March. The SLD phase in Spain, effective 15th of March, corresponds to a "state of alarm" declared by the Spanish government on the 13th, which implied a general closure of all retail commerce and a limitation on citizens' circulation rights across the country [46]. The FLD phase began on the 30th of March, during which all non-essential activity was banned and non-essential workers were ordered to stay at home, and lasted until the 9th of May [47].

In France, the WFH period considered here ranges from 9th to 16th of March and includes a ban on large gatherings as well as recommendations from companies to implement teleworking and limit their employees' business trips [48]. France went into an SLD on the 17th of March, which resulted in a closure of schools and institutions of higher education, ban on religious gathering, suspension of sport events, closure of cultural institutions, with only essential public services entitled to remain open [49]. The FLD period began on the 23rd of March when emergency measures were adopted, which saw severe restrictions on its citizens' movement and closure of business activity [50,51].

*2.2. Collection of Raw Data*

All the data used for the analysis in this paper were obtained from the ENTSO-E transparency platform [52]. This is the "central collection and publication of electricity generation, transportation and consumption data and information for the pan-European market". The following procedure lays down the steps involved in obtaining the load data for all countries for each phase, on weekdays and weekends.

- The daily load data for each country is available in the "Total Load-Day Ahead/Actual" data view option under the "Load" section of the menu bar on the transparency platform.
- The "country" option was selected, which displays the daily national total load. The latter includes both the Day-ahead total load forecast (in MW), as well as the Actual total load (in MW) for every country selected from the "Area" drop-down list on this page.
- The load data are given with an hourly resolution for all countries of interest, except for Germany, for which the data are available in a quarter-hourly resolution.
- Selecting the day for which the load was to be downloaded from the "Day" drop-down list and the "Total load- Day Ahead/Actual (CSV)" option from the "Export Data" drop down list leads to the export of the data as csv files.
- The latter contain the three columns (with twenty-four rows each): time (hourly resolution; quarter-hourly for Germany), day-ahead total load forecast (MW), and actual total load (MW).

Table A1 in Appendix A lists the time ranges for which the daily load data was downloaded for each phase of each country on weekdays and weekends. These dates reflect the exact time-frame during which the measures have been effective, as identified

and described in the subsection above. Figure 2 gives an overview of the timeline of the measures adopted and the energy demand for each of those periods.

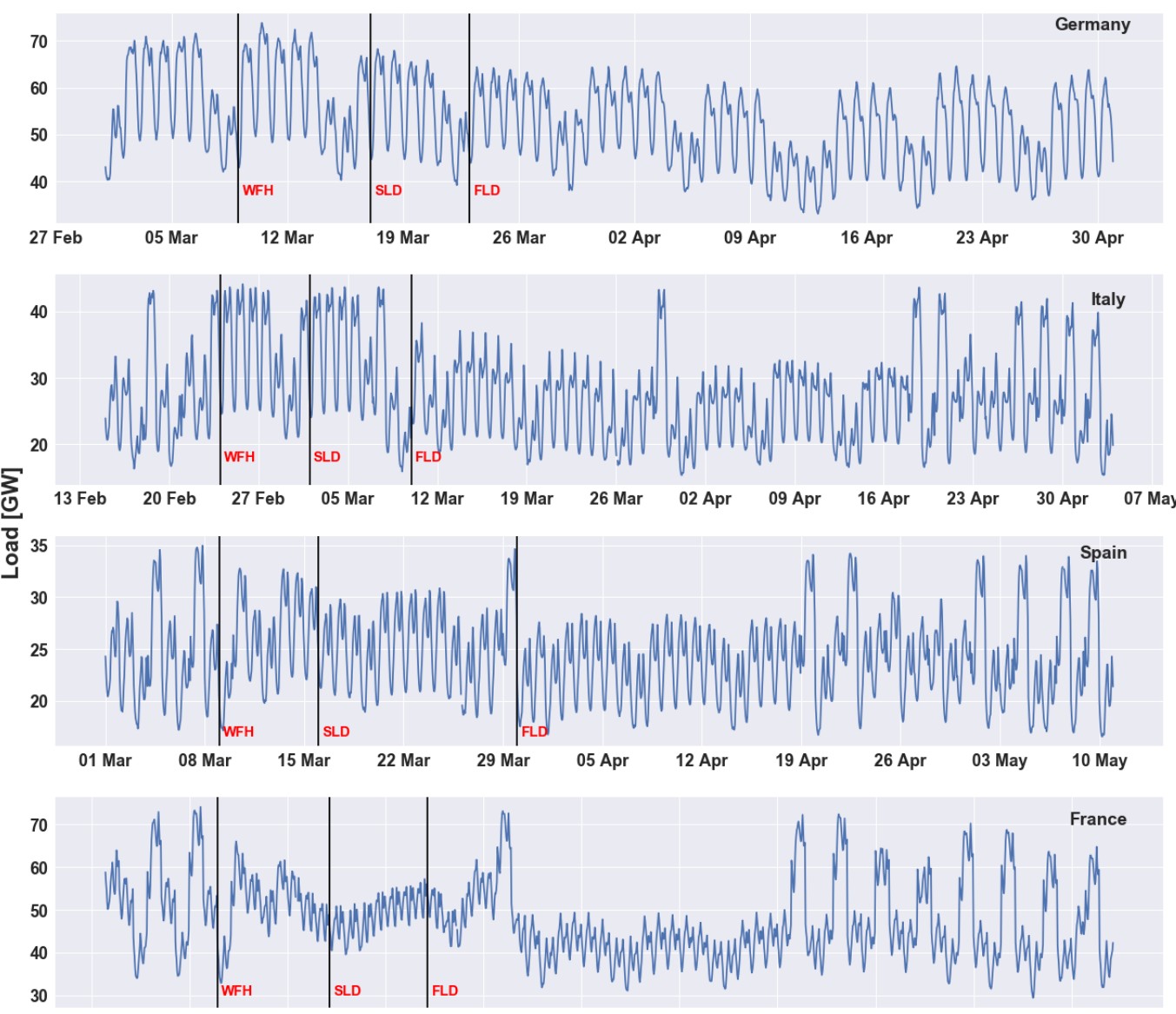

**Figure 2.** Timeline of the WFH, SLD, and FLD phases for Germany, Italy, Spain, and France (as denoted by the legends) with their corresponding energy demands in MW. The vertical black lines indicate the starting date of a given phase (labelled in red, adjacent to the line). For reference, see Table A1 for the starting date of each phase for each country.

### 2.3. Choice of Reference Period

In terms of choosing the reference period, the same procedure described above was followed for obtaining data for the reference load. The data for the reference profiles contain the load data of specific time ranges chosen from the previous five years, i.e., 2015 through 2019, from which the most representative reference load profile was generated.

The purpose of the reference load for each category is two-fold:

a. to normalise the corresponding daily demand curve of each period to the peak load value of its load profile,

b. to serve as comparison to determine the amount by which the demand changed, as a response to the three categories of measures adopted.

Since we consider the measures from the months of February through May, we needed to account for the usual seasonal change in daily demand. Hence, instead of a single reference load for all three phases, we considered using a separate reference load for each of the WFH, SLD, and FLD periods. However, owing to the short duration of the WFH and SLD weekday periods for all countries except for Spain (whose SLD period lasted for two work-weeks), one reference period common to the two and a separate reference for the FLD were sufficient to generate the representative reference profile. Thus, we chose two reference profiles for weekdays for all countries: one for WFH and SLD ("aggregated reference") and another for the FLD ("individual reference"), with the exception of Spain, which has three references due to a longer SLD phase, making it one reference for each category. As for weekends, one reference was chosen for each period.

For the aforementioned "individual references", the time range selected for all five years was exactly the same duration as that of each phase in 2020. That is, for instance, if WFH for weekends lasted from 14th March to 15th March (third weekend of March) in Germany in 2020, the data needed for generating the individual reference demand profile would be that for the third weekend of March from 2015 to 2019.

On the contrary, the time period for generating the "aggregated reference" was chosen to be the fortnight immediately preceding the weeks of WFH and SLD. This is illustrated by the following: if WFH for weekdays lasted from 10th March to 16th March and SLD from 17th March to 20th March in Germany in 2020, the aggregated reference profile would be generated from the load data from the two workweeks prior to these periods, from 2015 to 2019, e.g., 22nd February to 7th March in 2019. This time range is the closest period prior to the exact duration of the phases that does not overlap with either the WFH or SLD phases. This choice of the time range ensures that the aggregated reference is suited to represent the effects of the WFH and SLD phases equally and without favouring one period over the other, i.e., without biasing the choice towards representing one particular period more than the other. The exact dates for which the daily load data were obtained from the past five years for generating the reference load profile, for both weekdays and weekends, are tabulated in Table A2.

In order to validate the choice of using the load data from the past five years to obtain the reference period of each phase and the corresponding reference load profiles, the change in the annual electricity load as compared to the previous year, was investigated for all countries. Table A3 in Appendix A contains the average yearly change of the electricity demand (in percentage), of Germany, Italy, Spain, and France with respect to the previous year. The values were obtained from the "Supply, transformation and consumption of electricity" database under the "Energy statistics-quantities, annual data" option from Eurostat [53]. Data from 2015 onwards show that there has been no well-established trend for an increase or decrease in electricity consumption in Europe. This lack of an apparent trend or significant changes in the yearly consumption data justifies the suitability of using load data from 2015 to 2019 as they are, in order to arrive at reference load profiles, without having to adjust it for the change in demand or including a correction factor.

### 2.4. Data Cleaning and Processing

Section 2.4.1 through Section 2.4.5 explain the methodology followed to generate the datasets and subsequently plot them. In Section 2.4.1, we list the steps followed to clean the downloaded raw data, following which we detail the implementation of medoid-clustering to generate the representative dataset in Section 2.4.2. In Section 2.4.3, we explain how the daily load profiles were plotted from the aforementioned dataset, while in Section 2.4.4, we describe how the percentage- decrease datasets were obtained. Finally, we explain the plotting of the percentage change of demand, in Section 2.4.5.

All datasets were generated using code developed in Python. The Python code script is included in the supplementary material for this paper.

Following are the versions of Python and its packages which were used for the analysis: Python: 3.7.1; pandas: 0.23.4; numpy: 1.15.4; scipy: 1.1.0.

2.4.1. Processing Raw Data

After data collection, the first step towards producing a consolidated dataset of the daily loads of all countries under each measure was to clean the data files to a suitable format, so that they are ready to be manipulated further. This was accomplished by using a function that takes the path of the folder containing all data files (of .csv format) of a particular period as the argument and does the following:

- concatenates said data files into a data-frame (a labelled data structure and an object of the Pandas [54] library of Python, which loads datasets);
- extracts the starting time from the "Time (CET)" column of the data-frame
- removes the "Day-ahead Total Load Forecast (MW)" column;
- converts the extracted starting time into datetime format from string format;
- sorts the data by time;
- additionally, for Germany: the data is resampled from quarter-hourly to hourly, which is achieved using the DataFrame.resample('H').mean() function;
- returns the updated data-frame.

The data-frame returned by the function for each argument was stored separately in different data-frames—one for each measure per country and their corresponding references, for both weekdays and weekends.

2.4.2. The "Representative" Data-Frame

This data-frame was populated by the daily load profiles for each period of measure and their corresponding references. The daily load profiles are a series of hourly load values, normalized to the peak reference load, which in our assay was required to be the most representative load of each period at each hour of the day. Here, we describe the method for deriving this representative load.

The load recorded at each hour of the day across all the days constituting a period of measure were clustered together, and the medoid was chosen as the most representative load of the cluster (see Appendix B for the validation of the medoid as the most representative point). A medoid is a robust measure of central tendency and has the smallest average dissimilarity to all other data points in a cluster. Furthermore, it is restricted to always be a member of the data set. Thus, the most representative day of a period of measure (and which is what eventually inhabits the data-frame) is a series of twenty-four medoid values. This is illustrated with an example: the load values that make it to the "Germany-WFH" column in the consolidated data-frame are the medoid values of clusters at every hour; therefore, the resulting load at 08:00 h is the medoid of the cluster formed by loads recorded at 08:00 h of all days in the WFH period, i.e., 10th March to 16th March 2020.

This was implemented by simultaneously treating each category of measure through a loop with an index running 0 through 23, representing the hour of day. Within each iteration (i.e., for every hour of the day), the following steps are executed:

- A cluster is formed of load values recorded at that hour that matches the index of the loop, across all days of a period, for all phases for all countries, along with their references. This leads to the formation of five clusters for each of the countries: three of the measures and two for the references (and six for Spain, given three references as well). This was achieved by importing only those load-readings from the corresponding data-frames they are stored in, whose "hour" values of the data-frames' index matched with the loop's index
- Once the clusters are created, their medoids are computed and their values exported to their corresponding columns of the consolidated data-frame. Each row, corresponding to an hour of the day, is filled at the end of every iteration.
- The medoids were computed as the numpy.argmin [55] of the smallest absolute difference of the data points of a cluster to the median of the cluster, owing to the one-dimensionality of the data housed in the clusters. Additionally, they were cross-checked with the usual formulation of a medoid: as the argmin of the column sum or

the row sum of a pairwise distance matrix of all the data points of a cluster. The distance matrix was built as a square matrix using scipy's [56] squareform function and the pdist function from the scipy.spatial.distance module.

- At the end of twenty-four iterations, the consolidated data-frame contains the medoid load values of every hour of the day. Each column of the data-frame corresponds to the daily load profile of every measure per country and all the references.
- To obtain the normalised daily load profiles, the maximum load values of the reference profiles were identified and the hourly medoid load values in the consolidated data-frame were divided by the peak loads of their corresponding references. That is, for instance, Germany's WFH and SLD profiles were divided by the peak load of the "Germany_Reference1" profile, while the FLD profile was divided by the maximum load of the "Germany_Reference2" profile.

An identical processing was performed for the weekend data, with a separate data-frame and the only difference being the existence of three references—one corresponding to each of the phases.

### 2.4.3. Plotting Daily Demand Profile

Each of the columns housing the normalised daily load profiles in the updated and consolidated representative data-frame were plotted against every hour of the day. The reference profiles were plotted as dashed lines and the WFH, SLD, and FLD profiles were plotted, respectively, in green, yellow, and red as solid lines. The figure was generated using the "Seaborn" plotting style and has four subplots, one per country. An identical procedure was followed to generate the daily demand profile for the weekends as well.

### 2.4.4. The Percentage-Decrease Data-Frame

A second type of data-frame was generated from the aforementioned consolidated representative data-frame. This data-frame captures the change in demand, as a percentage, at every hour of the day for all the periods of measures, with respect to their corresponding references. The following steps were followed in order to generate this data-frame:

- Import all the contents of the consolidated data-frame into this percentage-decrease data-frame;
- Modify the twelve columns of the consolidated data-frame that contain the normalised loads of each phase and per country, such that these updated columns contain the change in demand. These are obtained in the following manner:
    - For every instance, i.e. for every hour of the day, compute (Column load − corresponding reference load)/corresponding reference load ∗ 100%;
- Remove the columns containing the normalised reference demand at every hour, for all the phases and all countries.

### 2.4.5. Plotting Percentage Change

Each of the columns housing the relative changes in demand in the percentage-change data-frame were plotted against every hour of the day. The resulting plot essentially collapses the daily load profile data into one figure: it has three subplots, each depicting the extent of the change in demand for all countries for WFH, SLD, and FLD, respectively. An identical procedure was followed to generate the change in demand as a percentage for the weekends as well.

### 3. Data Description

The datasets are stored in .csv format in an online repository  for ease of use in any statistical software [57]. Four final datasets were produced that represent the decrease in electricity demand under the three categories of measures adopted by the governments, respectively, (Figures 3 and 4): two datasets for weekdays and two for weekends.

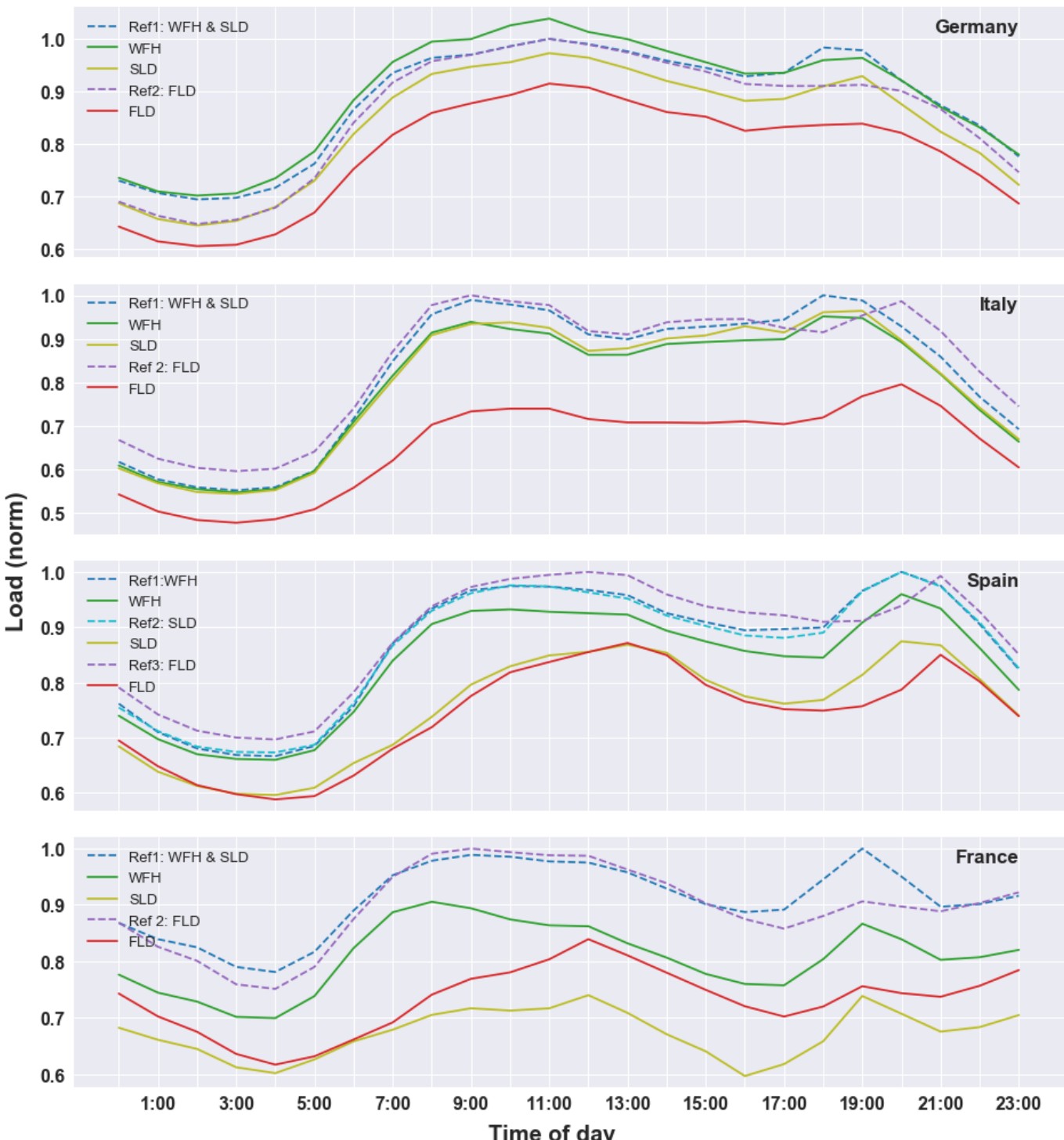

**Figure 3.** Weekday demand profiles during the WFH, SLD, and FLD periods along with their corresponding reference profiles for Germany, Italy, Spain, and France, respectively. The solid lines demonstrate the representative hourly demand for each period, normalised to the peak demand of its corresponding reference profile, which are shown as dashed lines. (See Tables A1 and A2 for the exact dates and durations of each phase for each country.)

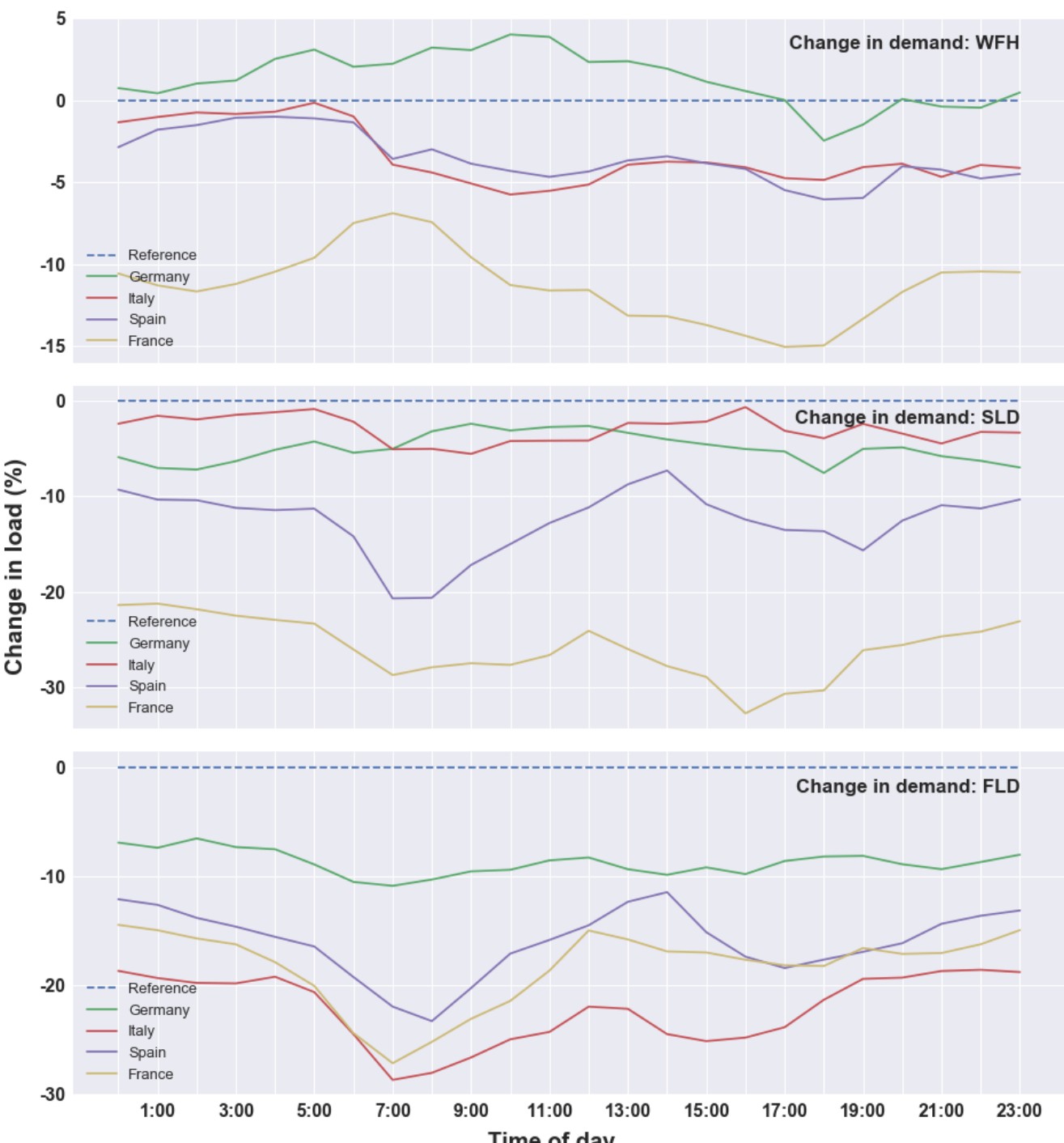

**Figure 4.** Systematization of the change in electricity demand of Germany, Italy, Spain, and France for the defined periods of WFH, SLD, and FLD, respectively, for weekdays. The solid lines depict the change, in percentage, in the national electricity demand with respect to their corresponding reference. (For reference, see Tables A1 and A2 for the exact dates and durations of each phase for each country.)

Table 1 provides the description of the entries in all columns in the files "weekday_normalised_load_profiles.csv" and "weekend_normalised_load_profiles.csv". These files contain the values of the per-hour-normalized demand for Germany, Italy, Spain, and France for the WFH, SLD, and FLD periods and for their corresponding reference periods, for weekdays and weekends, respectively.



**Table 1.** Description of the contents of the published datasets [weekday_normalised_load_ profiles.csv] and [weekend_normalised_load_profiles.csv]. The demands mentioned in the description column of the table are all normalized.

| Column Name | Description Index: Every Hour of the Day |
|---|---|
| DE_Ref1 | Demand during the reference period for WFH and SLD for Germany |
| DE_WFH | Demand of Germany under WFH |
| DE_SLD | Demand of Germany under SLD |
| DE_Ref2 | Demand during the reference period for FLD for Germany |
| DE_FLD | Demand of Germany under FLD |
| IT_Ref1 | Demand during the reference period for WFH and SLD for |
| IT_WFH | Demand of Italy under WFH |
| IT_SLD | Demand of Italy under SLD |
| IT_Ref2 | Demand during the reference period for FLD for Italy |
| IT_FLD | Demand of Italy under FLD |
| ES_Ref1 | Demand during the reference period for WFH for Spain |
| ES_WFH | Demand of Spain under WFH |
| ES_Ref2 | Demand during the reference period for SLD for Spain |
| ES_SLD | Demand of Spain under SLD |
| ES_Ref3 | Demand during the reference period for FLD for Spain |
| ES_FLD | Demand of Spain under FLD |
| FR_Ref1 | Demand during the reference period for WFH and SLD for France |
| FR_WFH | Demand of France under WFH |
| FR_SLD | Demand of France under SLD |
| FR_Ref2 | Demand during the reference period for FLD for France |
| FR_FLD | Demand of France under FLD |

Table 2 provides the description of the entries in all columns in the files "weekday_change_in_demand.csv" and "weekend_change_in_demand.csv". These files contain values for the percentage-change in power demand from the corresponding reference period at every hour of the day, for the WFH, SLD, and FLD periods for each country, for weekdays and weekends, respectively.

**Table 2.** Description of the contents of the published datasets [weekday_change_in_demand.csv] and [weekend_change_in_demand.csv].

| Column Name | Description |
|---|---|
|  | Index: every hour of the day |
| DE_WFH | % change of demand wrt reference for WFH, Germany |
| DE_SLD | % change of demand wrt reference for SLD, Germany |
| DE_FLD | % change of demand wrt reference for FLD, Germany |
| IT_WFH | % change of demand wrt reference for WFH, Italy |
| IT_SLD | % change of demand wrt reference for SLD, Italy |
| IT_FLD | % change of demand wrt reference for FLD, Italy |
| ES_WFH | % change of demand wrt reference for WFH, Spain |
| ES_SLD | % change of demand wrt reference for SLD, Spain |
| ES_FLD | % change of demand wrt reference for FLD, Spain |
| FR_WFH | % change of demand wrt reference for WFH, France |
| FR_SLD | % change of demand wrt reference for SLD, France |
| FR_FLD | % change of demand wrt reference for FLD, France |

*Intermediate Datasets*

The datasets "weekday_load_profiles.csv" and "weekend_load_profiles.csv" contain the pre-normalised loads (in MW) for all periods for all countries, for weekdays and weekends, respectively. That is, the load data at each hour of the day, per measure per country, is the medoid value obtained, as described in the Methods section. The contents of these datasets are exactly as described in Table 1, with the only exception that the load is not normalised. "weekday_load_profiles.csv" is used to generate "week-

day_normalised_load_profiles.csv" by normalising the load for each period to the maximum load of its corresponding reference period, and "weekday_change_in_demand.csv", by computing the percentage change in load data for each period with respect to the reference data for the corresponding period, for each hour. Similarly, "weekend_load_profiles.csv" generates "weekend_normalised_load_profiles.csv" and "weekend_change_in_demand.csv".

## 4. Usage Notes

Energy system models aid the planning of the energy transition of nations. These models use a wide variety of inputs such as cost projections of technologies, renewable resource profiles, and energy demand profiles [58]. Our datasets address the latter and provide two new contributions.

We provide electricity demand profiles at an hourly resolution for scenarios that show a strong remote-working behaviour as a first step toward a digitalized society. Concretely, the corresponding datasets are the green series in Figure 3 and the upper panel in Figure 4 and can be found in the DE_WFH, IT_WFH, ES_WFH and FR_WFH columns of the "weekday_normalised_load_profiles.csv" and "weekend_normalised_load_profiles.csv" datasets.

The second contribution comprises the demand profiles resulting from full lockdowns than can be used for assessing the resilience of power systems. These serve as a proxy for a minimum state of health of the system to be achieved in the recovery phase of power systems affected by extreme events like a pandemic. Concretely, these datasets are the red series in Figure 3 and the lower panel in Figure 4, and can be found in in the DE_FLD, IT_FLD, ES_FLD, and FR_FLD columns of the "weekday_normalised_load_profiles.csv" and "weekend_normalised_load_profiles.csv" datasets.

Overall, in this work, we categorise the COVID-19 restrictions adopted in Europe's four nations with the largest electricity consumption into phases of varying degrees of stringency and derive daily load profiles that are representative of each of these phases. We also systematise the change in electricity demand for each phase in each country. The analysis provided here is valuable for efficient energy management and load shedding in blackout situations. It is also valuable for balancing needs in future energy systems and making bids in such energy markets, where a greater share of remote working is expected. Our analysis is easily transferable to other countries and situations with energy consumption patterns similar to the ones analyzed here. The impact of this work is directed towards designing resilient energy system and accounting for extreme events and situations.

## 5. Discussion

The obtained curves on remote working are based on the currently best available empirical data. When validating the data, the heterogeneity of the durations of the same category across various countries becomes apparent, as well as of the definitions of the period of measures themselves, given the subjective nature of the definitions (as can be seen from the Methods section). For instance, the SLD period lasted for ten days in Spain, but only for four days in Germany. This might limit the comparability of the results for different countries.

The trends in the demand profiles display some inconsistencies, such as the WFH period in Germany having a higher demand than the reference case, the SLD period in Italy having a higher demand than the WFH period, and the FLD period in France having a higher demand than the SLD period.

In order to address these inconsistencies, it would be crucial to acknowledge the impact of a nation's response time to implement the measures announced: the time it takes for industries, companies, businesses, and the society as a whole to adapt to the changes and implement the measures, from the date of said measures having become effective. Furthermore, in all countries, the WFH measure, for instance, lasted for five to six days, which may be quite short a duration before a majority of the citizens make the shift to

working remotely and fully comply with the measures imposed. Hence, local factors like the weather or holidays can have a greater influence on the representative demand profile, resulting in a limited representation of this phase. Thus, the full extent of the impact of the shift may not be captured by or reflected in the energy demand during such short periods where the data is rather limited.

When comparing the results for different countries, one also needs to take into consideration various other factors such a country's work culture, its dominant industry type, and its existing infrastructures to support the digital shift, which could potentially affect the aforementioned response time. The effects of these factors may also be superimposed on the patterns of energy consumption visible during the periods that the measures lasted. For instance, reference [59] captures Europe's under-preparedness to work digitally: Germany and Italy lag behind most countries of the EU, owing to manufacturing being the dominant industry in Germany, poor access to ultra-fast fiber broadband in Italy, and a large proportion of workers never having previously worked remotely in both countries. Reference [60] finds that working remotely is feasible for only 56 percent of the German workforce. These factors could possibly have influenced Italy's and Germany's response time to the measures adopted. Even the mere increase in industrial production can serve as an influential pointer to address the aforementioned inconsistencies in the behaviour of the electricity demands.

**Supplementary Materials:** The following are available online at https://www.mdpi.com/article/10.3390/data6070076/s1. The Python code script is included in the supplementary material for this paper.

**Author Contributions:** Conceptualization, J.H.; methodology, S.M. and M.Y.; formal analysis, S.M.; investigation, S.M. and M.Y.; data curation, S.M.; writing—original draft preparation, S.M.; writing—review and editing, M.Y., J.H., H.-C.G., and M.F.; visualization, S.M.; supervision, J.H., H.-C.G., and M.F.; project administration, H.-C.G.; funding acquisition, H.-C.G. All authors have read and agreed to the published version of the manuscript.

**Funding:** This research was partially funded within the ReMo-Digital project supported by the German Federal Ministry of Economic Affairs and Energy (grant number FKZ BMWi 03EI1020B). The work was also supported in part by the Deutschlandstipendium from the German Federal Ministry of Education and Research.

**Data Availability Statement:** The data presented in this study are available in FigShare at https://doi.org/10.6084/m9.figshare.c.5513850.v1, accessed on 15 July 2021.

**Acknowledgments:** The authors would like to thank Ulrich Frey for a fruitful discussion on statistical methods for the technical validation. We would also like to thank Benjamin Fuchs for their counsel on data management. We are grateful to B.L. Radhakrishna for reviewing the code.

**Conflicts of Interest:** The authors declare no conflict of interest. The funders had no role in the design of the study; in the collection, analyses, or interpretation of data; in the writing of the manuscript; or in the decision to publish the results.

## Abbreviations

The following abbreviations are used in this manuscript:

| | |
|---|---|
| WFH | Work From Home |
| SLD | Soft Lockdown |
| FLD | Full Lockdown |

## Appendix A.

**Table A1.** Time ranges of the downloaded load data from ENTSO-E, for weekdays and weekends in 2020.

| Country | Period | Dates | |
|---|---|---|---|
| | | **Weekdays** | **Weekends** |
| Germany | WFH | 10 March–16 March | 14 March –15 March |
| | SLD | 17 March–20 March | 21 March–22 March |
| | FLD | 23 March–30 April | 28 March–26 April |
| Italy | WFH | 24 February–28 February | 29 February–01 March |
| | SLD | 02 March–09 March | 07 March–08 March |
| | FLD | 10 March–01 May | 14 March–03 May |
| Spain | WFH | 09 March–13 March | 07 March–15 March |
| | SLD | 16 March–27 March | 21 March–29 March |
| | FLD | 30 March–08 May | 04 April–10 May |
| France | WFH | 09 March–16 March | 14 March–15 March |
| | SLD | 17 March–23 March | 21 March–22 March |
| | FLD | 24 March–11 May | 04 April–10 May |

**Table A2.** Time ranges of the downloaded load data from ENTSO-E for generating the reference load profiles, for weekdays and weekends.

| Country | Phase | Dates | | | | | | | | | |
|---|---|---|---|---|---|---|---|---|---|---|---|
| | | **Weekdays** | | | | | **Weekends** | | | | |
| | | 2015 | 2016 | 2017 | 2018 | 2019 | 2015 | 2016 | 2017 | 2018 | 2019 |
| Germany | WFH | 23 Feb–6 Mar | 23 Feb–7 Mar | 23 Feb–7 Mar | 24 Feb–7 Mar | 22 Feb–7 Mar | 14 Mar–15 Mar | 12 Mar–13 Mar | 11 Mar–12 Mar | 17 Mar–18 Mar | 16 Mar–17 Mar |
| | SLD | | | | | | 21 Mar–22 Mar | 19 Mar–20 Mar | 18 Mar–19 Mar | 24 Mar–25 Mar | 23 Mar–24 Mar |
| | FLD | 25 Mar–30 Apr | 23 Mar–29 Apr | 23 Mar–28 Apr | 22 Mar–30 Apr | 21 Mar–30 Apr | 28 Mar–26 Apr | 26 Mar–24 Apr | 25 Mar–23 Apr | 31 Mar–28 Apr | 30 Mar–29 Apr |
| Italy | WFH | 9 Feb–20 Feb | 15 Feb–26 Feb | 13 Feb–24 Feb | 12 Feb–23 Feb | 11 Feb–22 Feb | 28 Feb–1 Mar | 27 Feb–28 Feb | 25 Feb–26 Feb | 24 Feb–25 Feb | 23 Feb–24 Feb |
| | SLD | | | | | | 7 Mar–8 Mar | 5 Mar–6 Mar | 4 Mar–5 Mar | 3 Mar–4 Mar | 2 Mar–4 Mar |
| | FLD | 10 Mar–1 May | 10 Mar–29 Apr | 10 Mar–1 May | 12 Mar–1 May | 11 Mar–1 May | 21 Mar–3 May | 26 Mar–1 May | 25 Mar–7 May | 25 Mar–6 May | 23 Mar–5 May |
| Spain | WFH | 24 Feb–9 Mar | 24 Feb–8 Mar | 23 Feb–8 Mar | 23 Feb–8 Mar | 25 Feb–8 Mar | 7 Mar–15 Mar | 5 Mar–13 Mar | 4 Mar–12 Mar | 10 Mar–18 Mar | 9 Mar–17 Mar |
| | SLD | 2 Mar–13 Mar | 1 Mar–14 1Mar | 1 Mar–14 Mar | 1 Mar–14 Mar | 1 Mar–14 Mar | 21 Mar–29 Mar | 19 Mar–27 Mar | 18 Mar–26 Mar | 24 Mar–1 Apr | 23 Mar–31 Mar |
| | FLD | 30 Mar–8 May | 30 Mar–6 May | 30 Mar–8 May | 30 Mar–8 May | 1 Apr–8 May | 4 Apr–10 May | 2 Apr–8 May | 1 Apr–7 May | 7 Apr–13 May | 6 Apr–12 May |
| France | WFH | 23 Feb–6 Mar | 23 Feb–7 Mar | 22 Feb–7 Mar | 22 Feb–7 Mar | 22 Feb–7 Mar | 14 Mar–15 Mar | 12 Mar–13 Mar | 11 Mar–12 Mar | 17 Mar–18 Mar | 16 Mar–17 Mar |
| | SLD | | | | | | 21 Mar–22 Mar | 19 Mar–20 Mar | 18 Mar–19 Mar | 24 Mar–25 Mar | 23 Mar–24 Mar |
| | FLD | 24 Mar–11 May | 24 Mar–11 May | 24 Mar–11 May | 26 Mar–11 May | 24 Mar–10 May | 28 Mar–10 May | 26 Mar–8 May | 25 Mar–7 May | 31 Mar–13 May | 29 Mar–12 May |

**Table A3.** Change of annual load for each country compared to the previous year, as a percentage.

| Year | Change of Load from Previous Year, % | | | |
|------|---------|-------|-------|--------|
|      | **Germany** | **Italy** | **Spain** | **France** |
| 2019 | −1.33 | −0.67 | −1.67 | −0.76 |
| 2018 | −0.14 | 0.51 | −0.14 | −0.69 |
| 2017 | 0.14 | 2.11 | 2.51 | −0.53 |
| 2016 | 0.41 | −0.57 | 0.78 | 1.45 |

**Appendix B. Justification and Validation of the Medoid Method**

*Appendix B.1. Medoid as the Most Representative Point*

A medoid returns the argument of the minimum of a sum of general pairwise dissimilarities. Therefore, by definition, it satisfies the condition of being the point having the smallest average distance to all other points in the cluster. Most importantly, it returns a non-synthetic observation; i.e., it is always one of the existing data points.

The medoid was chosen as the most representative point as it satisfies two conditions:

- it is the most centrally located point in a cluster or dataset, defined as that data point from the cluster, whose average dissimilarity to all other data points is minimal;
- it is robust to potential outliers in the dataset.

Despite being more computationally intensive than other central measures, such as the mean, the medoid is a robust measure, making it immune to outliers [61]. Qualitatively speaking, this implies that the medoid output would remain meaningful even if a fraction of the dataset were to be drastically altered. These properties of a medoid account for its suitability over other measures of central tendency such as the mean. Figure A1 visually captures these properties of the medoid by plotting the medoid profile along with the recorded profiles for every day of each of the WFH, SLD and FLD categories for weekdays in Germany.

*Appendix B.2. Validating Medoid Output*

In order to validate the load profile from the model output, i.e. the medoid load value at every hour of the day for each period, it was compared against the recorded load data for a randomly selected day in each period. For each phase, this evaluation was performed with both the model profile and the randomly selected profile normalised to the peak load of the randomly selected day, for each country and for weekdays and weekends, respectively.

To quantify the exact deviation of the model output from the randomly selected recorded load profile, the root mean square (RMS) differences between them were computed for several trials, i.e. for different random selections under each phase). These values are recorded in Tables A4 and A5 for weekdays and weekends, respectively. The fact that the RMS deviation between the two are very small (of the order of 0.001 to 0.1) implies that the medoid values closely represent the recorded data, thereby validating the method of usage of medoids. Furthermore, it can be seen from Figure A1 that the model profiles follow or mirror the actual profiles, which satisfies the criterion for a qualitative validation: the highest and lowest demands are observed at identical hours, for instance.

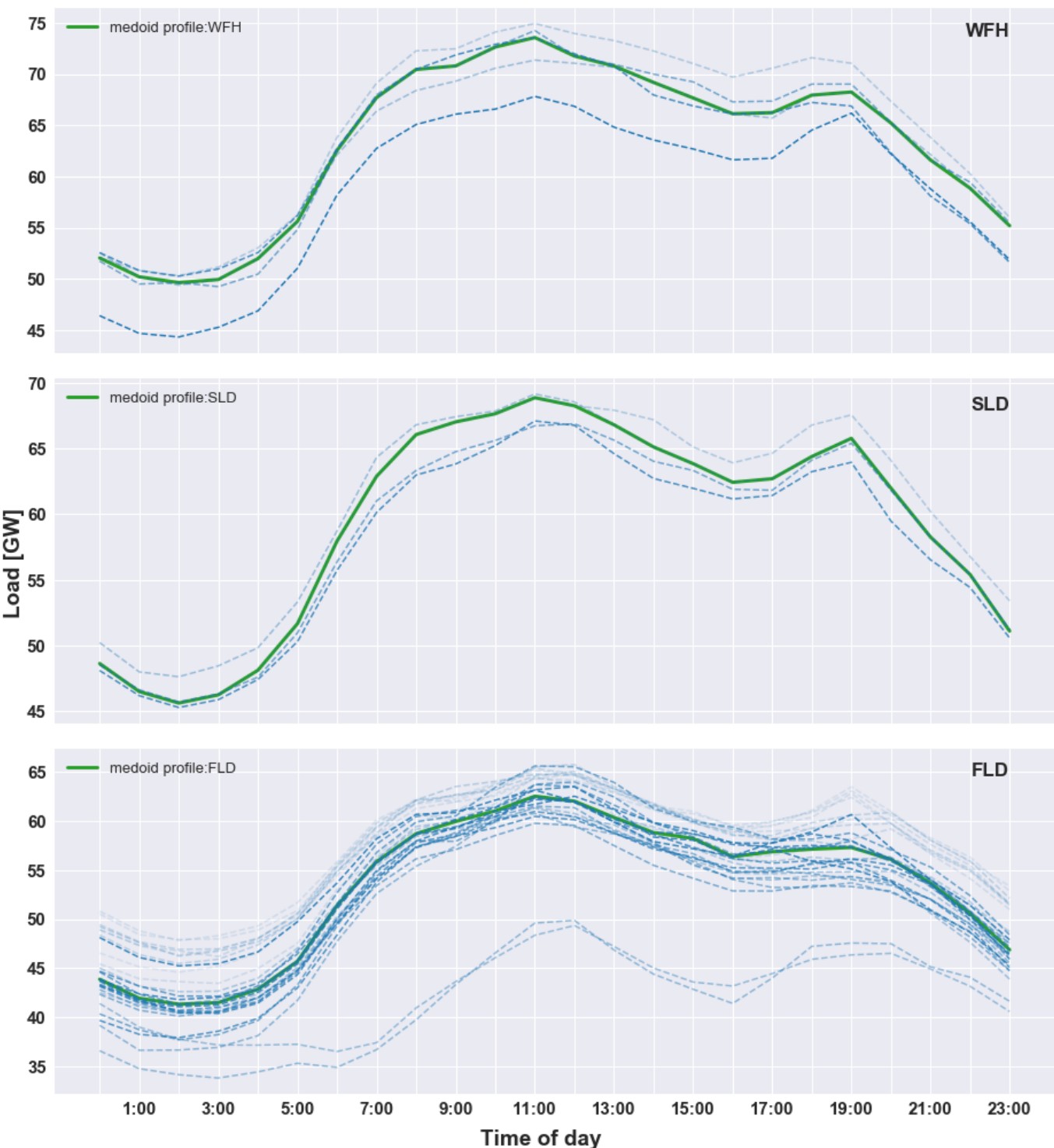

**Figure A1.** Medoid profile as the most representative of recorded profiles of all days for each phase. The solid green curve represents the medoid profile and the dashed blue curves of varying opacity represent the recorded profiles of every weekday in each of the WFH, SLD, and FLD phases in Germany.

**Table A4.** RMS differences between the model output and randomly-chosen recorded data for weekdays, for each phase for each country. The various trials correspond to different random selections.

| Trial | Measure | Germany | Italy | Spain | France |
|---|---|---|---|---|---|
| 1 | WFH | 0.00972 | 0.01253 | 0.02663 | 0.05764 |
| | SLD | 0.01415 | 0.01319 | 0.00361 | 0.01336 |
| | FLD | 0.05884 | 0.01361 | 0.02699 | 0.00952 |
| 2 | WFH | 0.02909 | 0.01581 | 0.0048 | 0.03189 |
| | SLD | 0.02708 | 0.01614 | 0.00596 | 0.01999 |
| | FLD | 0.01174 | 0.05764 | 0.01942 | 0.01773 |
| 3 | WFH | 0.01242 | 0.01357 | 0.02663 | 0.00254 |
| | SLD | 0.01815 | 0.00984 | 0.0078 | 0.01999 |
| | FLD | 0.01734 | 0.04033 | 0.01624 | 0.02813 |

**Table A5.** RMS differences between the model output and randomly chosen recorded data for weekends, for each phase for each country. The various trials correspond to different random selections.

| Trial | Measure | Germany | Italy | Spain | France |
|---|---|---|---|---|---|
| 1 | WFH | 0 | 0.09612 | 0.03822 | 0.0833 |
| | SLD | 0.073 | 0.11282 | 0.0441 | 0.01862 |
| | FLD | 0.04254 | 0.07235 | 0.00755 | 0.01048 |
| 2 | WFH | 0.08015 | 0.09612 | 0.05328 | 0.0833 |
| | SLD | 0.073 | 0 | 0.02872 | 0 |
| | FLD | 0.08564 | 0.01844 | 0.02672 | 0.04984 |

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
