# Peer review of "Impact of COVID-19 on Electricity Demand: Deriving Minimum States of System Health for Studies on Resilience"

_data, 2020_

Round 1
Reviewer 1 Report
Overall, I found this paper and the data it describes to be really interesting. I do think this information could be extremely useful for thinking about the design of resilient energy systems. The data seems original and the sources and methods are all well described and detailed, to me the appropriate quality controls were taken. I see that the dataset is available through Figshare, which is readily and easily accessible.
Although I think the methods and dataset itself are well described with much detail, in my opinion, the introduction and conclusion could be better framed and more detailed to really emphasize the importance of this type of data. I would like to see more discussion about how this data on minimum states of service during COVID could be used to inform say things like load shedding during blackouts or more efficient energy management, or perhaps something about priority loads for different sectors or populations within a nation. I felt that these issues were hinted at without much discussion or context. The conclusion could also be improved by bringing the article full circle, and revisiting how the dataset, after showing the results provides insights into the importance concepts discussed in the introduction. I also think it would help to add in the year, when talking about the timeline narratives during the Pandemic. Currently only the month (i.e., March) is referred too. In a few years, someone reading this may need to be reminded that we are talking specifically about 2020.
Author Response
General comments
Overall, I found this paper and the data it describes to be really interesting. I do think this information could be extremely useful for thinking about the design of resilient energy systems. The data seems original and the sources and methods are all well described and detailed, to me the appropriate quality controls were taken. I see that the dataset is available through Figshare, which is readily and easily accessible.
Thank you for your time and effort to review our work, and the constructive comments.
Comment 1
Although I think the methods and dataset itself are well described with much detail, in my opinion, the introduction and conclusion could be better framed and more detailed to really emphasize the importance of this type of data.
We have included some new references to similar works, and mentioned how our work adds value to and fills the gap in the existing literature (paragraphs 3 and 4, Introduction). Also, we further highlighted the additional value of our data sets in the second-last paragraph of the introduction. We follow up on these points in the Usage Notes section.
Comment 2
I would like to see more discussion about how this data on minimum states of service during COVID could be used to inform say things like load shedding during blackouts or more efficient energy management, or perhaps something about priority loads for different sectors or populations within a nation. I felt that these issues were hinted at without much discussion or context.
Thank you for these insights. We have included how our datasets can be valuable for load shedding and efficient energy management during blackouts, in the penultimate paragraph of the Introduction section, as “The amount and profile of electricity demand in the minimum state of system service can be used to derive tailored options for load shedding in blackout situations and for more efficient energy management.” This has also been mentioned in the concluding paragraph of the Usage Notes section.
Comment 3
The conclusion could also be improved by bringing the article full circle, and revisiting how the dataset, after showing the results, provides insights into the importance concepts discussed in the introduction.
We have enhanced the Usage Notes section by adding a final paragraph which further elaborates on the possible usage of our data. This aligns our goals, methods and results.
Comment 4
I also think it would help to add in the year, when talking about the timeline narratives during the Pandemic. Currently only the month (i.e., March) is referred too. In a few years, someone reading this may need to be reminded that we are talking specifically about 2020.
Good point! We incorporated this wherever it was missing or hard to catch. In the Methods, where the dates appear too often, we now try to establish the context clearly as 2020 in the introductory part of section 2.1.2: “The focus of this study is the time period from February to May 2020.” and additionally also in section 2.1.1 and the first paragraph of Methods. The caption to Figure 1 now includes “Definition and classification of categories for the restrictions taken to prevent the spread of COVID-19 in spring-2020.”
Reviewer 2 Report
This paper is about investigating the impact of the adopted COVID-19 measures on the electricity demand. The load data of three phases of increasing stringency for four major electricity-consuming countries of Europe are processed. It is a practical problem, while this paper does not write clearly. It looks like a project report instead of an academic paper. The reviewer has some questions:
- The influence of COVID-19 on electricity demand is well understood. To the best of my knowledge, there are many papers have been published in this area as follows:
“Bahmanyar A, Estebsari A, Ernst D. The impact of different COVID-19 containment measures on electricity consumption in Europe. Energy Research & Social Science. 2020 Oct 1;68:101683.”
“Abu-Rayash A, Dincer I. Analysis of the electricity demand trends amidst the COVID-19 coronavirus pandemic. Energy Research & Social Science. 2020 Oct 1;68:101682.”
“Alasali F, Nusair K, Alhmoud L, Zarour E. Impact of the COVID-19 Pandemic on Electricity Demand and Load Forecasting. Sustainability. 2021 Jan;13(3):1435.”
Therefore, the author must revise the introduction very carefully to provide a comprehensive state of the art in which to highlight their contribution.
- As the authors mentioned in the title of the paper, the minimum states of system health for studies on resilience can be derived. However, the reviewer did not find the relative explanations of how to find the minimum states of system health for studies on resilience.
- In order to illustrate the impact of the COVID-19 on the electricity demand, the selection of the aggregated reference profile is particularly important. The authors should explain why February 22nd- March 7th in 2019 are chosen in this paper.
- In figure 4, the load of Germany during WFH phase is higher than the aggregated reference profile, which is rarely happened in other countries or phases. What is the specific reason?
- There are some clerical errors in the paper. The authors are required to check the manuscript carefully before submission. The quality of the picture needs to be improved.
Author Response
General comments
This paper is about investigating the impact of the adopted COVID-19 measures on the electricity demand. The load data of three phases of increasing stringency for four major electricity-consuming countries of Europe are processed. It is a practical problem, while this paper does not write clearly. It looks like a project report instead of an academic paper. The reviewer has some questions:
We thank the reviewer for their time and effort to review our work, and for addressing some critical issues.
Comment 1
The influence of COVID-19 on electricity demand is well understood. To the best of my knowledge, there are many papers have been published in this area as follows:
“Bahmanyar A, Estebsari A, Ernst D. The impact of different COVID-19 containment measures on electricity consumption in Europe. Energy Research & Social Science. 2020 Oct 1;68:101683.”
“Abu-Rayash A, Dincer I. Analysis of the electricity demand trends amidst the COVID-19 coronavirus pandemic. Energy Research & Social Science. 2020 Oct 1;68:101682.”
“Alasali F, Nusair K, Alhmoud L, Zarour E. Impact of the COVID-19 Pandemic on Electricity Demand and Load Forecasting. Sustainability. 2021 Jan;13(3):1435.”
Therefore, the author must revise the introduction very carefully to provide a comprehensive state of the art in which to highlight their contribution.
We thank the reviewer very much for pointing at the recent papers on the topic. We included these and other recent publications in the third paragraph of the Introduction section. On this basis, we have also specified in more detail the added value of the data we provide and addressed how we aim to fill the gaps in the existing literature, in the fourth paragraph of the Introduction.
Comment 2
As the authors mentioned in the title of the paper, the minimum states of system health for studies on resilience can be derived. However, the reviewer did not find the relative explanations of how to find the minimum states of system health for studies on resilience.
We have added a qualitative description of the minimum state of system health in Section 2.1.1. There, we define the full lock down (FLD) as the state in which critical industries and infrastructures are still operational, while all non-essential businesses are closed. The Usage Notes section also highlights that the profiles derived under FLD serve as a proxy to the minimum state of system health and the first state to be achieved in the recovery phase. These profiles are the red series in Figure 3 and the lower panel of Figure 4.
Comment 3
In order to illustrate the impact of the COVID-19 on the electricity demand, the selection of the aggregated reference profile is particularly important. The authors should explain why February 22nd- March 7th in 2019 are chosen in this paper.
The period of February 22 to March 7 in 2019 was only used to illustrate the choice of time period selected for generating the aggregated reference, for the specific case of Germany. In 2018, the period selected was February 24 to March 7; February 23 to Mar 7 in 2017 and so on. These time ranges for all phases for all countries, for both weekdays and weekends, can be found in table A2 in Appendix A. The reason for selecting this phase is explained in the same paragraph which introduces the aggregated references. Choosing the fortnight before the combined period of WFH and SLD as the reference period ensures that it is close enough to the time range of WFH and SLD combined to have the same seasonal effects as the latter, and which also doesn’t overlap with either the WFH or SLD phases. This is explained in the paper as, “This time range is the closest period prior to the exact duration of the phases which doesn’t overlap with either the WFH or SLD phases. This choice of the time range ensures that the aggregated reference is suited to represent the effects of the WFH and SLD phases equally and without favouring one period over the other, i.e., without biasing the choice towards representing one particular period more than the other.”
Comment 4
In figure 4, the load of Germany during WFH phase is higher than the aggregated reference profile, which is rarely happened in other countries or phases. What is the specific reason?
This, along with other such visible inconsistencies are listed in the Discussion section. A complex combination of reasons are possible to address them. These factors are detailed in this section. While it is difficult to point at a specific reason for Germany, one possible reason is that the WFH period lasted for only 4 days, and there was no clear imposition of WFH either. As a result, it could’ve just been a transitory phase. Also, in such small intervals, local factors like weather have a larger influence on the demand profile. This has been added in the Discussion section as: “Also, in a time interval as short as this, the data is rather limited. Hence, local factors like the weather or holidays can have a greater influence on the representative demand profile, resulting in a limited representation of this phase.” in the second paragraph.
Comment 5
There are some clerical errors in the paper. The authors are required to check the manuscript carefully before submission. The quality of the picture needs to be improved.
Thank you. We have again checked the entire manuscript carefully, eliminating all errors we found. While the figures appear clear in our manuscript, we have anyway increased the resolution of the figures.
Reviewer 3 Report
The topic is important and interesting. I have some suggestions on the paper.
Comment 1. Abstract, highlight some key findings/results of the research work.
Comment 2. Section 1. Introduction, please address:
(a) Literature review should be enhanced. It is expected to share a summary on the methodology, results, and limitations of the existing works.
(b) Add an extra paragraph at the end to summarize what contents are going to be presented in coming sections.
Comment 3. Section 2. Methods, please address:
(a) Figure 1, regarding the definitions of some terms, do the definitions widely adopted? Please justify with citations.
(b) Authors mentioned “While Germany never explicitly issued a work-from-home…”, please keep using the abbreviation when the term has been abbreviated.
(c) How do 2.1 and 2.2 relate to the methodology? It seems that authors should reorganize 2.1 and 2.2 into a new section named background information and definitions.
(d) Elaborate the data collection process.
(e) Figure 2, the starting dates of WFH, SLD, and FLD should be indicated. Also, what are the periods for WFH, SLD, and FLD?
(f) Regarding the choice of reference period, does the reference period fixed, or it is user-defined? It seems that authors have shared the dataset in which the reference period is supposed to be fixed.
(g) Is there missing data?
Comment 4. Section 3. Data Description, please address:
(a) Please clarify “[see Figshare link on the title page ]”.
(b) Figures 3 and 4, please specify the dates/periods?
(c) What is the granularity (sampling frequency) of the data?
Comment 5. Section 4. Usage notes, please address:
(a) Share some research topics.
Author Response
General comments
The topic is important and interesting. I have some suggestions on the paper.
Thank you for your time and effort to review our work, the constructive and specific comments, and the kind wording.
We have tried to highlight the reduction in demand and change in shape of load profile in the abstract as “The analysis could unravel the influence of the different measures to the energy consumption and the differences among the four countries. It is observed that the daily peak load is considerably flattened and the total electricity consumption decreases by up to 30% under the circumstances brought about by the COVID-19 restriction.”
Comment 2. Section 1. Introduction, please address:
(a) Literature review should be enhanced. It is expected to share a summary on the methodology, results, and limitations of the existing works.
Additional paragraphs detailing the current state of research and identifying the research gaps in the literature are now included (paragraphs 3 and 4, Introduction). We also mention how our work aims to close these research gaps.
(b) Add an extra paragraph at the end to summarize what contents are going to be presented in the coming sections.
Sure, this has now been included as the last paragraph of the Introduction section.
Comment 3. Section 2. Methods, please address:
(a) Figure 1, regarding the definitions of some terms, do the definitions widely adopted? Please justify with citations.
No, these are not widely adopted but rather defined by us, as mentioned in the introduction section and in section 2.1.1.
(b) Authors mentioned “While Germany never explicitly issued a work-from-home…”, please keep using the abbreviation when the term has been abbreviated.
Thank you. We have replaced the complete terms (Work-from-home, etc.) with abbreviations (WFH, SLD, etc) throughout the main text in the manuscript. However, we left them in their expanded form in figure-labels and conclusions (in the Usage Notes section) so that they are more comprehensible, especially in the first view for readers who may only skim the paper.
(c) How do 2.1 and 2.2 relate to the methodology? It seems that authors should reorganize 2.1 and 2.2 into a new section named background information and definitions.
Good point! Defining the categories of restrictions and classifying specific measures implemented into these categories serves as the first step in identifying the dates. This has been described and now included under a section “Background info and definitions” in Methods.
(d) Elaborate the data collection process.
While we feel that we have already included all the steps we used to collect the data for our analysis, we did convert the paragraph into bullet-points to offer more clarity.
(e) Figure 2, the starting dates of WFH, SLD, and FLD should be indicated. Also, what are the periods for WFH, SLD, and FLD?
As we tried to include the starting dates of each phase for each country, the figure appeared very dominated by text. Instead, to preserve the clarity of the contents in the figure, we have linked table A1 in the caption. This table contains the periods of WFH, SLD and FLD, along with the starting dates of each phase.
(f) Regarding the choice of reference period, does the reference period fixed, or it is user-defined? It seems that authors have shared the dataset in which the reference period is supposed to be fixed.
The reference period is selected according to the duration that each phase lasted in 2020. It is not a fixed period. Owing to the short duration of WFH and SLD in all countries except Spain, we have defined an aggregated reference period for these two phases, and an individual reference for the FLD. The time range for the individual reference period is the same calendar days in previous years as it was in 2020, while that for the aggregated reference period is the fortnight before the calendar days in 2020, from the previous years. The chosen dates are tabulated in table A2 in Appendix A for generating the reference profile.
(g) Is there missing data?
The only source of the raw data is ENTSO-E , which can typically have some missing data, but for the considered time period for all the years and countries, no missing data was encountered.
Comment 4. Section 3. Data Description, please address:
(a) Please clarify “[see Figshare link on the title page ]”.
We have replaced it with a reference to the Figshare link containing the datasets.
(b) Figures 3 and 4, please specify the dates/periods?
Since each phase for each country lasts for different durations, we feel that it might be an overload of text if the dates are specified in the figures. Instead, we have linked tables A1 and A2 in the caption of these figures, for ease of referencing the dates.
(c) What is the granularity (sampling frequency) of the data?
The load data on ENTSO-E has an hourly resolution for all our countries considered, except for Germany, for which the load is available at a quarter-hourly resolution. This is mentioned in the list in section 2.2. Also, the resampling of load data from quarter-hourly to hourly, in the case of Germany, is described in section 2.4.1.
Comment 5. Section 4. Usage notes, please address:
(a) Share some research topics.
We have included an additional paragraph in the Usage Notes section which highlights the value of our work and future research directions. Some of these are: efficient energy management and load shedding in blackout situations, balancing needs in future energy systems, making bids in future energy markets where a greater share of remote working is expected, etc.
Round 2
Reviewer 3 Report
Authors have significantly improved the quality of the manuscript. I have some minor follow-up comments.
Comment 3. Section 2. Methods, please address:
(a) Figure 1, regarding the definitions of some terms, do the definitions widely adopted? Please justify with citations.
No, these are not widely adopted but rather defined by us, as mentioned in the introduction section and in section 2.1.1.
Follow-up comment: A better justification is needed if the terms are defined by authors. It should be in a way that the terms should share common characteristics with the terms adopted in literature.
Comment 5. Section 4. Usage notes, please address:
(a) Share some research topics.
We have included an additional paragraph in the Usage Notes section which highlights the value of our work and future research directions. Some of these are: efficient energy management and load shedding in blackout situations, balancing needs in future energy systems, making bids in future energy markets where a greater share of remote working is expected, etc.
Follow-up comment: It would be great if authors could share some references [x] for the research topics.
Author Response
General comments
Authors have significantly improved the quality of the manuscript. I have some minor follow-up comments.
Once again, thank you for your time and effort to review our revisions, as well as the kind wording.
Comment 3. Section 2. Methods, please address:
- Figure 1, regarding the definitions of some terms, do the definitions widely adopted? Please justify with citations.
Author response: No, these are not widely adopted but rather defined by us, as mentioned in the introduction section and in section 2.1.1.
Follow-up comment: A better justification is needed if the terms are defined by authors. It should be in a way that the terms should share common characteristics with the terms adopted in literature.
We have included an additional paragraph in section 2.1.1. highlighting the motivation for classifying specific measures into categories and defining them. Additionally, in the second paragraph of the same section where we define our categories, we have cited a few works in literature which adopt terms similar to ours, to draw parallels. We would like to note that no phases similar to our WFH definition were found in literature. Hence, we have elaborated on the WFH definition to include stay-at-home orders apart from just remote working, which better justifies the naming of the category.
Comment 5. Section 4. Usage notes, please address:
- Share some research topics.
Author response: We have included an additional paragraph in the Usage Notes section which highlights the value of our work and future research directions. Some of these are: efficient energy management and load shedding in blackout situations, balancing needs in future energy systems, making bids in future energy markets where a greater share of remote working is expected, etc.
Follow-up comment: It would be great if authors could share some references [x] for the research topics.
Thank you for the suggestion. We have included references for these research topics in the second- last paragraph of the Introduction section where we highlight the value of our datasets for the energy community.